EMBO
*reports*

*Scientific Report*

# Structural insight into recognition of phosphorylated threonine-4 of RNA polymerase II C-terminal domain by Rtt103p

Olga Jasnovidova[*], Magdalena Krejcikova, Karel Kubicek & Richard Stefl[**]

## Abstract

Phosphorylation patterns of the C-terminal domain (CTD) of largest subunit of RNA polymerase II (called the CTD code) orchestrate the recruitment of RNA processing and transcription factors. Recent studies showed that not only serines and tyrosines but also threonines of the CTD can be phosphorylated with a number of functional consequences, including the interaction with yeast transcription termination factor, Rtt103p. Here, we report the solution structure of the Rtt103p CTD-interacting domain (CID) bound to Thr4 phosphorylated CTD, a poorly understood letter of the CTD code. The structure reveals a direct recognition of the phospho-Thr4 mark by Rtt103p CID and extensive interactions involving residues from three repeats of the CTD heptad. Intriguingly, Rtt103p's CID binds equally well Thr4 and Ser2 phosphorylated CTD. A doubly phosphorylated CTD at Ser2 and Thr4 diminishes its binding affinity due to electrostatic repulsion. Our structural data suggest that the recruitment of a CID-containing CTD-binding factor may be coded by more than one letter of the CTD code.

**Keywords** NMR; RNA processing; RNAPII CTD code; structural biology
**Subject Categories** Post-translational Modifications, Proteolysis & Proteomics; Structural Biology; Transcription

## Introduction

RNA polymerase II (RNAPII) utilizes a long and flexible carboxyl-terminal domain (CTD) of its largest subunit to specifically recruit protein/RNA-binding factors during transcription [1–5]. The CTD consists of tandem repeats with conserved consensus Tyr1-Ser2-Pro3-Thr4-Ser5-Pro6-Ser7 that is repeated 26 times in yeast and 52 times in humans [6]. The CTD sequence is post-translationally phosphorylated at serines (Ser2, Ser5 and Ser7) and Tyr1 in a dynamic manner, yielding specific patterns that are recognized by appropriate factors in coordination with the transcription cycle events [3–5,7].

Additionally another highly conserved position, Thr4, was reported to be phosphorylated both in yeast and humans [8–12]. However, the levels of pThr4 in cells remain controversial based on two recent mass-spectrometry studies [10,11]. Substitution of Thr4 to Ala (T4A) or Val (T4V) is lethal for chicken and human cells [12–14]; however, the same mutants are viable in yeast [9,15,16]. In humans, genomewide studies revealed increasing levels of pThr4 throughout the gene body with the peak after the poly-A site [12]. In agreement with this, the T4A mutant showed defect in transcription elongation [12]. In yeast, the pThr4-mark is enriched along the whole gene body, similarly to the pTyr1-mark [17]. Both marks go down prior recruitment of transcription termination factors [17]. Therefore, it was suggested that the pThr4 mark along with the pTyr1-mark prevent binding of transcription termination factors during transcription elongation [17]. However, recent high resolution ChiP-nexus data suggested a different role for the pThr4 mark involved in transcription termination and post-transcriptional splicing [18].

It has been unclear for a long time what protein factors are recruited through the pThr4 signal. Interestingly, yeast transcription termination factor, Rtt103p, well known to be associated with the pSer2-mark [17,19,20], was identified as a part of the interactome of RNAPII phosphorylated at Thr4 [18]. Based on the overlay of NET-seq and ChiP-nexus profiles, Rtt103p coincides with the pThr4 mark after poly-A site. Both, deletion of the entire Rtt103p protein or expression of Rpb1 T4V CTD mutant, cause similar RNAPII pausing defect after poly-A site. The authors suggested a model, in which both pSer2 and pThr4 marks can contribute to the recruitment of Rtt103p to the poly-A site [18]. This concept is also supported by recent mass-spectrometry analyses of RNAPII CTD population pulled down by Rtt103p, which revealed simultaneous presence of pThr4 and pSer2 marks [11].

To understand the puzzling roles of the pSer2 and pThr4 marks in recruitment of transcription termination factor Rtt103p, we solved NMR structure of the pThr4 CTD peptide in complex with Rtt103p CTD-interacting domain (CID). Our structure reveals for the first time a direct readout of the pThr4 mark within the CTD. We also reveal significantly larger interaction area of Rtt103p with the CTD peptide than previously reported [20]. Next, we show that two adjacently positioned phosphorylations, pSer2 followed by pThr4, inhibit

CEITEC – Central European Institute of Technology, Masaryk University, Brno, Czech Republic
*Corresponding author. E-mail: olga.jasnovidova@ceitec.muni.cz
**Corresponding author. Tel: +420 549492436; E-mail: richard.stefl@ceitec.muni.cz

the binding of Rtt103p CID due to a charge–charge repulsion of the two closely positioned phosphate moieties. Finally, we propose that the CTD code is degenerated, as Rtt103p reads the pThr4 and pSer2 marks equally well using the same molecular mechanism.

## Results and Discussion

### Rtt103p CID binds equally well Thr4 and Ser2 phosphomarks

To test the binding affinity of Rtt103p CID towards pThr4-CTD *in vitro,* we performed an equilibrium-binding assay using fluorescence anisotropy (FA) (Fig 1B). The experiment revealed that Rtt103p binds pThr4-CTD with a $K_D$ of $15 \pm 1$ µM, which is 2.5 times weaker binding than to the CTD with the pSer2 mark ($K_D = 6.0 \pm 0.2$ µM). This finding is in a good agreement with previous co-immunoprecipitation studies, where Rtt103p was pulled down by RNAPII with the pThr4 mark and successfully competed out by pSer2-CTD or pThr4-CTD antisera [18]. Doubly phosphorylated pThr4-CTD at both Thr4 displayed increased binding affinity due to avidity effects ($K_D = 6 \pm 0.2$ µM). Remarkably, if the pSer2 and pThr4 marks are positioned adjacently, binding affinity ($K_D = 43 \pm 2$ µM) is lowered almost to the level of non-phosphorylated CTD ($K_D = 64 \pm 2$ µM). The pSer5 phosphorylation mark was

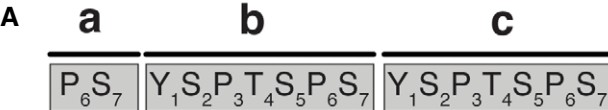

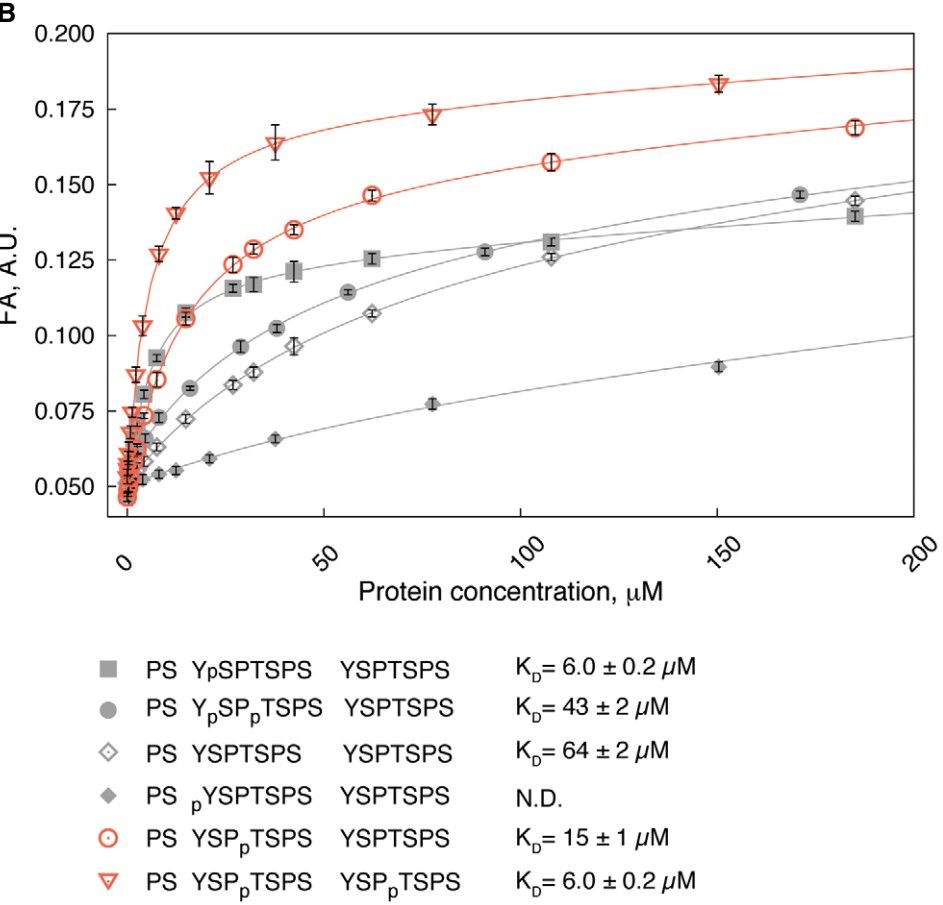

**Figure 1.  How CTD phosphorylations modulate binding to Rtt103p CID.**

A   Numbering of residues and order of heptad repeats of the CTD peptide used throughout the study.

B   Equilibrium binding of Rtt103p CID with fluorescently labelled CTD peptides monitored by fluorescence anisotropy (FA). Rtt103p CID titrated into 10 nM FAM-labelled CTD peptides. Peptide sequences, corresponding binding isotherms and dissociation constant ($K_D$, $\pm$ standard deviation of the fit) are shown. FAM, 5,6-carboxyfluorescein. N.D., not determined.

also previously shown to abolish and lower the binding with Rtt103p or its close human homologue [20,21]. Next, we introduced the pTyr1 mark to the central heptad of the CTD peptide, which completely abolished the binding with Rtt103p (Fig 1B). This suggests that $Y_{1b}$ is accommodated in the hydrophobic pocket following the previously established binding model for CIDs [17].

## NMR structure of Rtt103p CID bound to CTD with phospho-threonine mark

To reveal the structural basis of pThr4 recognition, we solved solution structure of a reconstituted complex that harbours Rtt103p CID (3-131) and a 16-amino acid peptide, pThr4-CTD (PS YSP(pT)SPS

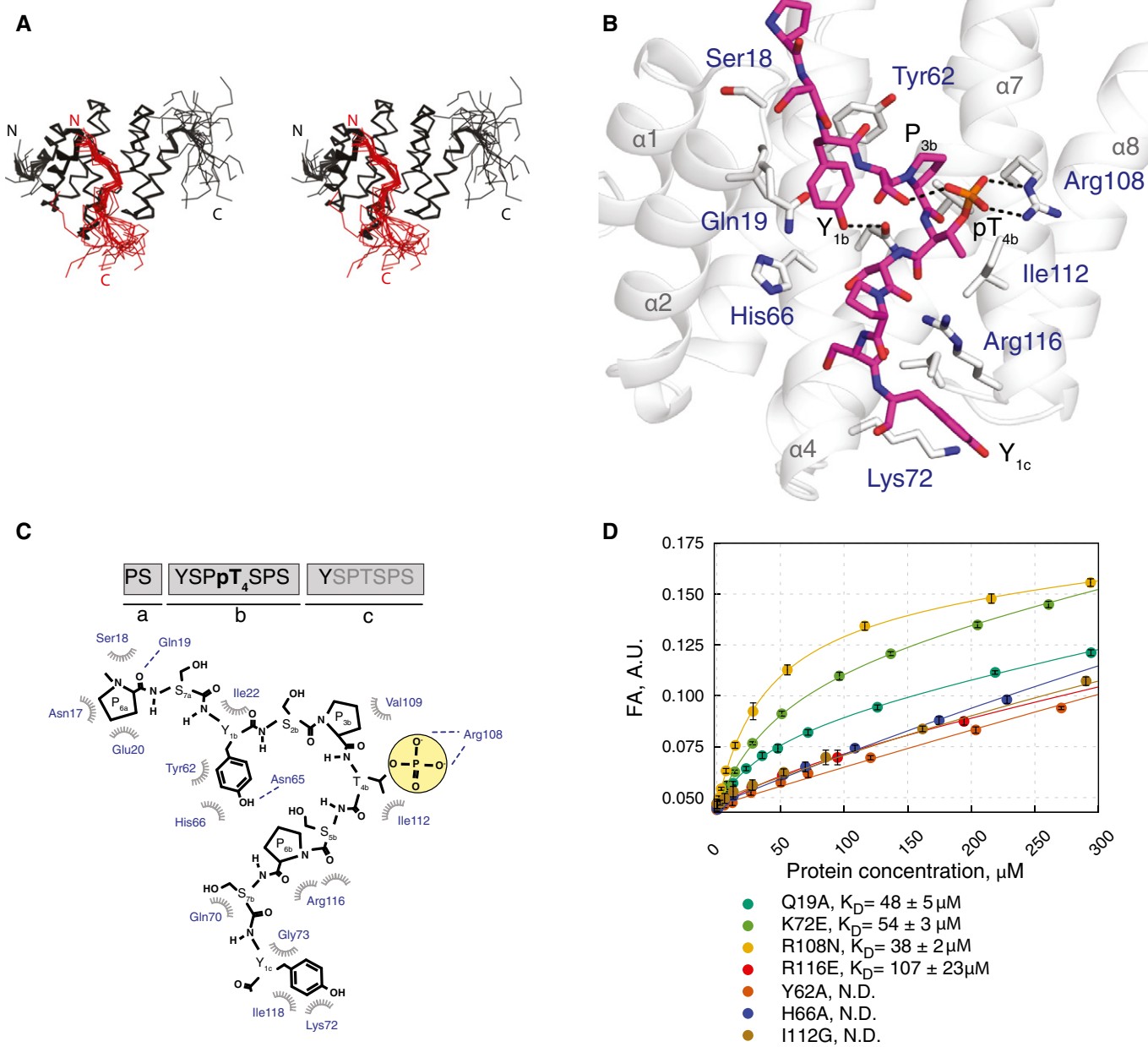

**Figure 2.    Solution structure of Rtt103p CID in complex with pThr4-CTD.**

A    Overlay of the 20 lowest energy structures of Rtt103p CID (black ribbon) complexed with pThr4-CTD (red ribbon) shown in stereo. N- and C-termini of the protein and peptide are indicated.

B    Solution structure of Rtt103p CID (grey helices) bound to the pThr4-CTD peptide (magenta sticks). Highlighted Rtt103p CID residues (grey sticks, blue labels) form hydrophobic contacts and putative hydrogen bonds (dashed black lines) with pThr4-CTD peptide.

C    Schematic diagram of Rtt103p CID (blue) and pThr4-CTD (black) interactions (hydrophobic contacts, spoked arcs; hydrogen bonds, dashed lines).

D    Equilibrium binding of Rtt103p CID mutants with pThr4-CTD peptide monitored by fluorescence anisotropy (FA). Rtt103p CID mutants titrated into 10 nM FAM-labelled CTD peptides. Corresponding binding isotherms and $K_D$s (± standard deviation of the fit) are shown. FAM, 5,6-carboxyfluorescein. N.D., not determined.

YSPTSPS; Fig 2A–C; Table 1). We used this peptide with a single phosphorylation to avoid binding in multiple registers that would complicate NMR data analyses. The resulted structure of Rtt103p CID is formed by eight α-helices in a right-handed superhelical

**Table 1. NMR and refinement statistics for the Rtt103p CID-pThr4 CTD complex.**

|  | Rtt103p CID–pThr4 CTD complex |
|---|---|
| **NMR distance & dihedral constraints** | |
| Distance restraints | |
| Total NOEs | 3,639 |
| Intra-residue | 843 |
| Inter-residue | 2,796 |
| Short | 1,691 |
| Medium | 1,104 |
| Long | 844 |
| Hydrogen bonds | 99 |
| Intermolecular distance restraints | 47 |
| Total dihedral angle restraints[a] | 198 |
| **Structure statistics[b]** | |
| Violations (mean and s.d.) | |
| Number of distance restraint violations > 0.5 Å | 0.10 ± 0.31 |
| Number of dihedral angle restraint violations > 15° | 13.8 ± 2.33 |
| Maximum dihedral angle restraint violation (°) | 39.57 ± 9.55 |
| Maximum distance constraint violation (Å) | 0.30 ± 0.12 |
| Deviations from idealized geometry[b] | |
| Bond lengths (Å) | 0.00355 ± 0.00008 |
| Bond angles (°) | 1.702 ± 0.012 |
| Average pairwise r.m.s.d. (Å)[b] | |
| Rtt103p CID (7–12; 19–31; 36–48; 54–73; 77–94; 100–116; 121–133) | |
| Heavy atoms | 0.91 ± 0.13 |
| Backbone atoms | 0.25 ± 0.05 |
| CTD (143–152) | |
| Heavy atoms | 2.06 ± 0.42 |
| Backbone atoms | 1.42 ± 0.34 |
| Complex | |
| All complex heavy atoms | 1.12 ± 0.16 |
| All complex backbone atoms | 0.61 ± 0.14 |
| Ramachandran plot statistics[c] | |
| Residues in most favoured regions (%) | 88.5 |
| Residues in additionally allowed regions (%) | 10.1 |
| Residues in generously allowed regions (%) | 0.9 |
| Residues in disallowed regions (%) | 0.5 |

[a]α-helical dihedral angle restraints imposed for the backbone based on the CSI.
[b]Calculated for an ensemble of the 20 lowest energy structures.
[c]Based on PROCHECK analysis [42].

arrangement (Fig 2A and B), out of which helices α2, α4 and α7 contact the pThr4-CTD peptide at residues $P_{6a}$, $S_{7a}$, $Y_{1b}$, $P_{3b}$, $pT_{4b}$, $S_{7b}$ and $Y_{1c}$ (Figs 2B and C, and EV1). This minimal CTD-binding moiety binds Rtt103p CID with a $K_D$ of 18 ± 1 μM (assayed by FA), which is almost identical as pThr4-CTD used for structural determination. The structure is similar to the one of Rtt103p CID–pSer2-CTD complex [20] in terms of the overall CID fold and the conformation of the N-terminal part of the CTD peptide, but entirely different for the C-terminal part of the CTD peptide (Figs 3, EV1, and EV2).

### Recognition of the phospho-threonine CTD by Rtt103p

The upstream part of the pThr4-CTD peptide adopts a β-turn conformation at $S_{2b}P_{3b}pT_{4b}S_{5b}$ and docks into a hydrophobic pocket of the Rtt103p CID that is formed by Ile22, Tyr62, His66, Val109 and Ile112, using $Y_{1b}$ and $P_{3b}$ residues (Fig 2B and C). The peptide conformation in the hydrophobic pocket is further stabilized by a hydrogen bond between hydroxyl of $Y_{1b}$ and the side-chain amide of Asn65. This hydrophobic pocket of Rtt103p is highly conserved, and mutations of residues Tyr62 and His66 (not affecting the structural integrity; Fig EV3) completely abolish the binding with pThr4-CTD (Figs 2D and EV4). $P_{3b}$ is inserted into the hydrophobic pocket next to Val109 and has a *trans* conformation of the $S_{2b}P_{3b}$ peptidyl-prolyl bond. As a result of this arrangement, both the $S_{2b}$ and $pT_{4b}$ side chains are positioned closely to each other in the solvent exposed area and form intramolecular hydrogen bond between the hydroxyl group of $S_{2b}$ and phospho-group of $T_{4b}$. The phospho-group of $pT_{4b}$ forms a hydrogen bond with the guanidinium group of Arg108. This is a critical interaction with the pThr4 mark, as confirmed by the affinity data for the Arg108Asn mutant (Fig 2D). Akin to Rtt103p, also other CID-containing proteins such as SCAF4/8 [22], RPRD1A/1B/2 [21] and CHERP [21] contain the equivalent arginine in the CID pocket. It will be interesting to see whether these human proteins really recognize pThr4-CTD as well and whether the pThr4 mark is relevant to their functions. Other CID-containing proteins in yeast, such as Nrd1p and Pcf11p, do not contain the equivalent arginine and these proteins were absent in the pThr4-CTD interactome [18].

Remarkably, we observed multiple strong intermolecular NOEs among the aromatics of $Y_{1c}$ in the downstream region of the CTD peptide and the C-terminal parts of helices α4 and α7 (Figs 2C, EV1, and EV5). The interaction of $Y_{1c}$ at the tip of helices α4 and α7 creates a second turn in the peptide at residues $pT_{4b}S_{5b}P_{6b}S_{7b}$, bringing two backbone carbonyl groups in close proximity and allows for their interaction with the guanidinium group of Arg116 (Figs 2B and C, and EV1). The side chain of $Y_{1c}$ forms numerous hydrophobic contacts with Lys72, Gly73, Ile118. Arg116Glu and Lys72Glu charge swapping mutants cause affinity drop of $K_D$ = 107 ± 23 μM and $K_D$ = 54 ± 3 μM, respectively (Fig 2D). The similar arrangement of the downstream region of the CTD was observed in the crystal structure of close human homologue of Rtt103p CID, RPRD1A, where the arginine forms a hydrogen bond with carbonyl of $T_{4b}$ and $P_{6b}$ [21]. The Arg116 position is conserved in RPRD1A/1B (Arg114) and RPRD2 (Arg130) (Fig EV4) [21]. Interestingly, the coordination of tyrosine from the third heptad repeat $Y_{1c}$ was not observed previously in the structure of Rtt103p CID bound to the CTD with Ser2 phosphorylation [20]. The previous study used a CTD peptide

lacking the complete binding moiety (PS YSPTSPS Y) that possibly precluded the accommodation of the downstream part of CTD peptide including the second tyrosine (Y$_{1c}$; Fig EV1). The comparison of chemical shift perturbations of Rtt103p upon binding to the singly phosphorylated pSer2-CTD and pThr4-CTD peptides with complete binding moiety suggests similar accommodation of downstream region of both peptides (Fig EV2).

### *Cis-trans* equilibrium of the Ser–Pro prolyl-peptidyl bond

We also tested as to whether two proximal phosphorylation marks (pSer2/pThr4) on the CTD peptide can alter the *cis-trans* equilibrium of the neighbouring prolyl-peptidyl bond (Fig EV6A). It has been shown that the *cis-trans* equilibrium of the CTD is critical for its recognition by cognate proteins [23–25] and the *trans* conformation of the Ser–Pro prolyl-peptidyl bond is required for the β-turn formation [20,23,26,27]. To exclude the possibility that a highly populated *cis* conformer would attenuate the binding of the CTD peptide with two phospho-marks, we assayed the conformational population of mono- and diphosphorylated peptides using the [$^1$H,$^{13}$C]-HSQC spectra of PS Y(pS)$^{13}$CP(pT)SPS YS and PS YS$^{13}$CP(pT)SPS YS peptides, where all P$_{3b}$ carbons were $^{13}$C-isotopically labelled (Fig EV6B). In case of pThr4-CTD, we observed 6.6% of the *cis* conformer. We obtained virtually identical number for the pSer2pThr4-CTD peptide, where the *cis* conformation was populated at 7.8%. Our data suggest that the double phosphorylation at pSer2/pThr4 of the CTD does not influence the ratio of *cis-trans* conformers. Next, we titrated the PS YS$^{13}$CP(pT)SPS YS peptide with Rtt103p CID and monitored the titration by [$^1$H,$^{13}$C]-HSQC experiment (Fig EV6C). The spectra show the disappearance of peaks that correspond to the *cis* conformation during titration, indicating a shift in the *cis-trans* equilibrium towards the *trans* conformation of the S$_{2b}$–P$_{3b}$ prolyl-peptidyl bond that is required for the β-turn formation. The peaks corresponding to the *trans* conformation of P$_{3b}$ moved upon titration with protein, reflecting the accommodation of the proline in the hydrophobic pocket of Rtt103p CID.

### CTD code degeneration

The complex of Rtt103p CID–pThr4-CTD reported here represents the first structure capturing the recognition of the CTD phosphorylated at threonine and explains the structural basis of why Rtt103p can be a part of the pSer2- and pThr4-CTD interactomes. Previous reports suggested that Thr4 phosphorylations could interfere with CTD binding by destabilizing the β-turn conformation that is required for CTD binding [17,26]. However, our structure shows that the pThr4 mark is directly recognized by Rtt103p and also that the phosphate group of pThr4 forms intramolecular hydrogen bond stabilizing the bound CTD conformation. This conformation involves the β-turn at S$_{2b}$P$_{3b}$pT$_{4b}$S$_{5b}$ that is a prerequisite for an effective docking into the hydrophobic pocket of Rtt103p CID (Figs 2B and C, and 3). Interestingly, the intramolecular hydrogen bond that stabilizes the β-turn mirrors the one of the Ser2 phosphorylated CTD bound to Rtt103p (Fig 3B). The Rtt103p Arg108Asn mutant has also a similar drop in affinity for pThr4-CTD and pSer2-CTD, $K_D = 38 \pm 2$ μM and $K_D = 44 \pm 2$ μM, respectively. These observations suggest that CTD modifications preventing intramolecular stabilization of the β-turn should negatively affect CTD binding. Indeed, we observed that doubly phosphorylated CTD at Ser2 and Thr4 binds to Rtt103p as weak as unmodified CTD (Fig 1). Electrostatic repulsion between closely arranged phosphates of pSer2 and pThr4 interfere with the formation of the bound CTD conformation and the peptide with pSer2/pThr4 marks cannot be accommodated in the binding pocket of Rtt103p (Fig 3). In support of this, the coexistence of the pSer2/pThr4 marks in the same repeat has not been detected by recent mass-spectrometry analysis of RNAPII CTD population pulled down by Rtt103p [11]. Our structure also explains lethality of the Thr4Glu CTD mutant in yeast [18]. Permanent substitution for glutamate mimics Thr4 phosphorylation that interferes with Ser2 phosphorylation, which consequently prevents binding of the CTD to cognate proteins as described above.

The individual letters of the CTD code have so far been associated with unique information translated to stimulation or inhibition of recruitment of CTD readers. Comparison of Rtt103p

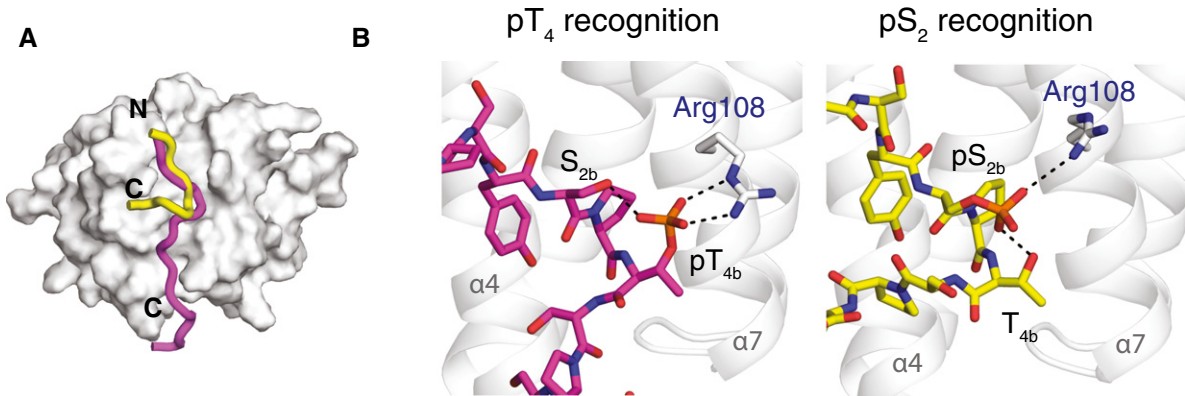

**Figure 3. Degeneracy of the CTD code.**

A  Superposition of pThr4-CTD (magenta; PDB ID: 5LVF) and pSer2-CTD (yellow; PDB ID: 2L0I) peptides on the Rtt103p CID surface (grey). N- and C-termini of the peptides are indicated.

B  Close view on the phospho-recognition site of Rtt103p CID. Interaction of pThr4- (magenta sticks, left) and pSer2-CTD (yellow sticks, right) peptides with Arg108 (grey). Hydrogen bonds of the phospho-groups are indicated with black dashed lines.

CID–pThr4-CTD structure with the Rtt103p CID–pSer2-CTD complex shows fascinating feature that the same interaction pocket of Rtt103p can read two different phosphorylation patterns of the CTD (pSer2 and pThr4) using the same mechanism and involves the same residues (mainly Arg108; Fig 3). Based on our structural findings, we suggest that the CTD code can be degenerated when read by CID-containing proteins. In other words, the recruitment of a single CTD-binding factor may be coded by more than one letter of the CTD code. As a consequence of this redundancy, CID-containing CTD-binding factors can be recruited to the poorly conserved heptad repeats of the CTD (e.g. the CTD of fruit fly) or they can tolerate some errors or imperfections in phosphorylation of the CTD [1,3,28].

# Materials and Methods

## Cloning and protein purification

pET28b-Rtt103p CID was a gift from B. Lunde [20]. Rtt103p CID point mutants were obtained by QuikChange site-directed mutagenesis kit (Stratagene). Resulting constructs were verified by DNA sequencing and then transformed into *E. coli* BL21-Codon Plus (DE3)-RIPL cells (Stratagene). Rtt103p CID (3-131–6xHIS) was expressed and purified as previously described [20].

## NMR measurements and structure determination

All NMR spectra for the backbone and side-chain assignments were recorded on Bruker AVANCE III HD 950, 850 and 700 MHz spectrometers equipped with cryoprobes at a sample temperature of 20°C using 1 mM uniformly $^{15}N,^{13}C$-labelled Rtt103p CID in 35 mM KH$_2$PO$_4$, 100 mM KCl, pH 6.8 (20°C) (90% H$_2$O/10% D$_2$O). Initial backbone resonance frequency assignment was transferred from BMRB entries 17044 and 16411 and confirmed by HNCA spectrum. The spectra were processed using TOPSPIN 3.2 (Bruker Biospin), and the protein resonances were assigned manually using Sparky software (Goddard T.G. and Kneller D.G., University of California, San Francisco). For the assignment of the side-chain proton and carbon resonances, 4D version of HCCH TOCSY [29] was measured with a non-uniform sampling. Acquired data were processed and analysed analogously as described previously [30,31].

All distance constraints were derived from the three-dimensional $^{15}N$- and $^{13}C$-edited NOESYs collected on a 950 MHz spectrometer. Additionally, intermolecular distance constraints were obtained from the three-dimensional $F_1$-$^{13}C$/$^{15}N$-filtered NOESY-[$^{13}C,^1H$]-HSQC experiment [32,33], with a mixing time of 150 ms on a 950 MHz spectrometer. The NOEs were semi-quantitatively classified based on their intensities in the 3D NOESY spectra. The initial structure determinations of the Rtt103p-CTD complex were performed with the automated NOE assignment module implemented in the CYANA 3.97 program [34]. Then, the CYANA-generated restraints along with manually assigned protein-CTD intermolecular restraints were used for further refinement of the preliminary structures with AMBER16 software [35]. These calculations employed a modified version (AMBER ff14SB) of the force field [36], using a protocol described previously [37,38]. The 20 lowest energy conformers were selected (out of 50 calculated) to

form the final ensemble of structures. The atomic coordinates for the NMR ensemble of the Rtt103p CID-pThr4-CTD complex have been deposited in the Protein Data Bank under ID code 5LVF and in Biological Magnetic Resonance Bank under ID code 34041. Molecular graphics were generated using PyMOL (The PyMOL Molecular Graphics System, Version 1.8 Schrödinger, LLC).

## Fluorescence anisotropy

The equilibrium binding of Rtt103p CID constructs to differently phosphorylated CTD was analysed by fluorescence anisotropy. The CTD peptides were N-terminally labelled with the 5,6-carboxyfluorescein (FAM). The measurements were conducted on a FluoroLog-3 spectrofluorometer (Horiba Jobin-Yvon Edison, NJ). The instrument was equipped with a thermostatted cell holder with a Neslab RTE7 water bath (Thermo Scientific). Samples were excited with vertically polarized light at 467 nm, and both vertical and horizontal emissions were recorded at 516 nm. All measurements were conducted at 10°C in 35 mM KH$_2$PO$_4$, 100 mM KCl (pH 6.8). Each data point is an average of three measurements. The experimental binding isotherms were analysed by DynaFit using 1:1 model with non-specific binding [39].

## *Cis-trans* population estimation

For the estimation of the *cis-trans* population of conformers around the Ser-Pro peptide bond, aliphatic [$^{13}C,^1H$]-HSQC was collected using 1 mM sample of peptide PSYS$^{13}C$P(pT)SPSYS or PSY(pS)$^{13}C$P (pT)SPSYS in 35 mM KH$_2$PO$_4$, 100 mM KCl (pH 6.8) in 90% H$_2$O/10% D$_2$O at 20°C on Bruker AVANCE III HD 700 MHz spectrometer. For the [$^{13}C,^1H$]-HSQC titration experiment, 0.2 mM PSYS$^{13}C$P(pT) SPSYS peptide was used and 1.2 mM Rtt103p CID stock was added. $^1H$-$^{13}C_\gamma$ and $^1H$-$^{13}C_\beta$ peaks were integrated using Sparky routine (Goddard T.G. and Kneller D.G., University of California, San Francisco). Population was estimated as a ratio of the peak volume of a given conformation to the sum of volumes of all conformations.

## Determination of chemical shift perturbation (CSP) value

Chemical shift perturbation (CSP) value is defined as the normalized length of a vector $E_j$, whose components are differences $\delta_{ji}$ between observed chemical shifts (bound form) and chemical shifts from a reference experiment (free form). Index $j$ represents the amino acid type within the primary sequence of the protein. Weight factors for each atom type $w_H = 1$ and $w_N = 0.15$ were used.

$$|E_j| = \sqrt{\sum_{i=H,N} w_i \delta_{ji}^2}$$

## Peptides used in the study

The following peptides were synthesized by JPT (Berlin, DE) and Clonestar (Brno, CZ): FAM-PSY(pS)PTSPSYSPTSPS; FAM-PSYSP(pT)SPSYSPTSPS; FAM-PSY(pS)P(pT)SPSYSPTSPS; FAM-PSYSPTSPSYSPTSPS; FAM-PSYSP(pT)SPSYSP(pT)SPS; FAM-PSYSP (pT)SPSYS; FAM-PS(pY)SPTSPSYSPTSPS; PSYSP(pT)SPSYSPTSPS; PSYS$^{13}C$P(pT)SPSYS; PS Y(pS)$^{13}C$P(pT)SPSYS.

**Expanded View** for this article is available online.

## Acknowledgements

We thank K. M. Harlen and L. S. Churchman for sharing preliminary results and fruitful discussion, B. Lunde for gift of pET28b-Rtt103p CID plasmid, T. Kabzinski for sharing pET28b-Rtt103p CID Y62A, H66A and I112G mutant plasmids, J. Novacek for discussion, P. Kuzmic and C. Hofr for helpful advice, M. Sebesta for critical reading of the manuscript. We acknowledge the Josef Dadok National NMR Centre, CEITEC—Masaryk University, supported by the CIISB research infrastructure (LM2015043 funded by MEYS CR) for their support with obtaining scientific data presented in this article. This project has received funding from the European Research Council (ERC) under the European Union's Horizon 2020 research and innovation programme (Grant Agreement No. 649030). This publication reflects only the author's view, and the Research Executive Agency is not responsible for any use that may be made of the information it contains. The results of this research have been acquired within CEITEC 2020 (LQ1601) project with financial contribution made by the Ministry of Education, Youths and Sports of the Czech Republic within special support paid from the National Programme for Sustainability II funds. This work was also supported by the Czech Science Foundation (13-18344S to R.S.; M.K. and K.K. were supported by 15-24117S).

## Author contributions

OJ designed the experiments, prepared protein and peptide samples, measured and analysed FA, assigned spectra, calculated and refined structure, and wrote the manuscript; MK collected and processed 4D HCCH TOCSY spectra, and assisted with structure refinement; KK collected and processed NMR spectra, and assisted with structure calculation and refinement; RS designed the experiments, assisted with structure calculation and refinement, and wrote the manuscript.

## Conflict of interest

The authors declare that they have no conflict of interest.

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
