## [Review Process File · EMBO Reports]

Manuscript EMBO-2016-43723

Structural insight into recognition of phosphorylated threonine-4 of RNA polymerase II C-terminal domain by Rtt103p

Olga Jasnovidova, Magdalena Krejcikova, Karel Kubicek, and Richard Stefl

Corresponding author: Richard Stefl, Masaryk University

Review timeline:

Submission date:	28 November 2016
Editorial Decision:	04 January 2017
Revision received:	09 March 2017
Editorial Decision:	21 March 2017
Revision received:	23 March 2017
Accepted:	29 March 2017

Editor: Esther Schnapp

Transaction Report:

1st Editorial Decision

04 January 2017

Thank you for the submission of your manuscript to our journal. We have now received the enclosed reports on it.

As you will see, all referees acknowledge that the findings are interesting and novel. They mainly suggest text changes, ask for a few more structural data, and remark that cross-reactivity of the Thr4-P and Ser2-P antibodies should be analyzed. Given the relatively minor comments, I suggest that all of them should be addressed.

We would thus like to invite you to revise the manuscript with the understanding that the referee concerns must be fully addressed and their suggestions taken on board. Please address all referee concerns in a complete point-by-point response. Acceptance of the manuscript will depend on a positive outcome of a second round of review. It is EMBO reports policy to allow a single round of revision only and acceptance or rejection of the manuscript will therefore depend on the completeness of your responses included in the next, final version of the manuscript.

Revised manuscripts should be submitted within three months of a request for revision; they will otherwise be treated as new submissions. Given the 3 main figures, I suggest that you publish the study as a short report. For short reports, the revised manuscript should not exceed 25,000 characters (including spaces but excluding materials & methods and references) and 5 main plus 5 expanded view figures. You could move one or two of the expanded view figures to the main manuscript file, if you want to. The results and discussion sections must further be combined, which will help to shorten the manuscript text by eliminating some redundancy that is inevitable when discussing the

same experiments twice.

REFEREE REPORTS

Referee #1:

The yeast transcription termination factor Rtt103p has a RNA polymerase II CTD interaction domain (CID) and can bind specifically to serine 2 and threonine 4 phosphorylated heptad repeats of CTD. While the interaction of Rtt103p with CTD phosphorylated at serine 2 residues has been studied before (Lunde et al., 2010, Ref. 20), it is currently unclear, which structural changes in CID can discriminate between Ser2P and Thr4P residues in CTD.

Stefl and coworker now provide evidence that only a minimal change in the structure of CID is responsible for this discrimination. Using NMR analysis the authors show that the arginine at position 108 can switch between two positions that contacts either the hydroxyl group of Ser2 and the phospho group of Thr4P, or, vice versa, the hydroxyl group of Thr4 and the phospho group of Ser2P. Consistent with this finding the authors show that interaction of Rtt103p interaction is blocked, if both residues are phosphorylated.

The NMR studies are complemented and confirmed by the study of Rtt103p and CTD mutants. The CTD Thr4/Glu mutant is lethal. This may be explained by the finding that the acidic glutamine residue blocks the binding of Rtt103p (and other factors) to Ser2P residues.

The authors discuss a model in which the 3' termination machinery can be recruited by different CTD phosphorylation marks and that the "CTD code" is degenerated. In an alternative (or more general) model, Ser2P and Thr4P marks may regulate termination of transcription in a gene specific manner. Testing of this model would require the analysis of CTD Thr4 mutants in transcription termination.

In summary, this well performed study explains how a CID can bind two different CTD marks just by switching the conformation of single amino acid in the binding domain.

Minor point: page 3 line 14: in this study a T4A mutant was analysed

Referee #2:

Phosphorylations on the C-terminal domain (CTD) of the largest subunit of RNA polymerase II play important roles in mediating the recruitment of RNA processing factors and chromatin modifiers during transcription. The work by Jasnovidova et al. presents an interesting new finding related to Threonine4-phosphorylation (Thr4-P). Based on Fluorescence Anisotropy and NMR structural studies, the authors clearly show that the CTD-interacting domain (CID) of Rtt103, a well-known CTD Ser2-P binder, can also bind to Thr4-P with comparable affinity *in vitro*. Interestingly, the phosphate on either Ser2 or Thr4 occupies a nearly identical position in the structure. In addition, they demonstrate that a CTD with phosphorylations on both Ser2 and Thr4 has decreased affinity for Rtt103 CID, consistent with the structure predicting that this doubly phosphorylated state would interfere with binding. However, recent mass spec papers suggest this double phosphorylation probably doesn't exist *in vivo*. Altogether, this study sets forth an important and intriguing model where proteins that specifically recognize Ser2-P might also bind to Thr4-P and vice versa.

Comments:

1. To argue that CID interaction with Ser2-P or Thr4-P is specific among singly phosphorylated forms, it would have been good to include another singly phosphorylated peptide in the FA experiment as a control, such as Ser7-P or Ser5-P (as done in Lunde et al. 2010; Ni et al. 2014). If the authors think that this has been established well enough in other papers that used shorter peptides, this point should at least be mentioned for clarification.

2. One concern I have about Figure 1 FA experiment is that the Thr4-P peptide signal is significantly higher than the other peptides and perhaps not even completely saturated. This could lead to an apparent K_d lower than the actual value. Can the authors comment on this?

2. It would be meaningful to add some discussion on the possibility of other CIDs (i.e. Pcf11, Nrd1, SCAF8) recognizing Thr4-P as well. As far as I understand, Pcf11 CID lacks the equivalent of Arg108 that serves as a hydrogen bond partner with Ser2-P or Thr4-P, while SCAF8 has Arg112. The authors' insights on this aspect might lead to and help followup studies trying to understand why Pcf11 might not have this characteristic and whether Thr4-P might be relevant to SCAF8 function.

3. Regarding the difference between this structure and that from Varani group, could the different pathway of the C-terminal CTD region also be affected by the fact that Varani used a peptide where both repeats were phosphorylated at Ser2-P? Their FA experiments also showed a higher affinity for two Ser2-P repeats versus one. Could that be because they were binding in two different registers? The Greenblatt group also reported binding between RPRD1A/B CID and a 'SPS YpSPTSPS YpSPTSPS' peptide. However, their CTD peptide was only visualized down to the second Tyr, perhaps also suggesting a second Ser2-P might affect interactions with CIDs.

4. p6. The paper says Arg108 is direct reader of Thr4-P, but this residue also makes a similar contact when the phosphate is on Ser2. The Asn mutation similarly blocks Ser2P peptide in FA assay. So I don't think it's accurate to say this residue is a direct reader of Thr4P, but rather only the phosphate at this location, whether or not it's on Thr4 or Ser2. The model shows two bonds between Thr4-P and Arg108, but only one with Ser2-P. Shouldn't that translate to a higher affinity for Thr4-P?

5. I would clarify two points in the discussion related to the recent mass spec papers from the Buratowski and Eick groups:

A. p4 and 9. Reference 11 (Suh et al) reports only a very small amount of Thr4P (far less than both Ser2P and Ser5P) in the Rtt103 pulldowns. The Schuller paper (reference 10) doesn't pull down Rtt103, but reports a higher level of Thr4P in total pol II, similar to Ser2P (although that paper also has a substantial False Location Rate). So it's unclear how much Thr4P exists in cells.

B. p5. Neither mass spec study finds very much Tyr1-P in yeast, so the proposed model (reference 17) where Tyr1-P blocks CID binding to an adjacent Ser2-P or Ser5-P is very unlikely.

6. The remarkably similar structures of the Thr4-P and Ser2-P peptides bound to Rtt103 makes me question whether the antibodies against these modifications are really as specific as advertised (the original Eick paper doesn't show any quantitation). It seems likely there might be at least some level of cross-reactivity. I wonder if the authors have tested their CTD peptides with the antibodies. Although this might seem slightly off topic, the results would shed light on whether this is a general issue that needs to be taken into consideration for all Ser2-P and Thr4-P experiments. Given how much people rely on the antibodies, this is an important issue for the CTD field.

Referee #3:

The manuscript of Jasnovidova et al reports the 'first' structure of transcription termination factor Rtt103p-CID bound to a singly Thr4-phosphorylated CTD diheptad N-terminally extended by Pro-Ser (pT4) using heteronuclear NMR spectroscopy. An increased binding affinity (> 4X) CTD for Rtt103p upon phosphorylation at the first repeat of T4 is determined by fluorescence anisotropy. The structure and binding affinities are compared with the recognition of S2 phosphorylated CTD peptides as reported by Lunde et al. The authors propose that the pT4 CTD peptide has an extended binding interface on Rtt103p compared to pS2 binding reported by Lunde et al, mainly due to the usage of longer peptides in the current study. Finally, the authors conclude that double S2 and T4 phosphorylation reduces the CID interaction due to electrostatic repulsion, while the affinities and molecular functions of single phosphorylation at S2 and T4 appear comparable and redundant. The experimental approaches and data are in general sound and the manuscript represents a short report on a structure with binding studies. Unfortunately, the discussion appears short and lacks rigor and depth, and will benefit from a more comprehensive analysis, especially regarding a comparison with the previous study on the recognition of S2 phosphorylation by the Rtt103p CID.

Specific comments:

- In the introduction the authors refer to the previous study by Mayer et al stating that pT4 prevents binding of transcription termination factors. In contrast, the present work and other studies describe controversially that the transcription termination factor Rtt103p is recruited more efficiently with

pT4 phosphorylation of the CTD. These controversial findings should be discussed.

- The authors argue that the previous study by Lunde et al exhibited 'incomplete binding' (bottom, p.4) due to the usage of a shorter CTD peptide. This is a speculation that should be clarified.

1. The authors should compare (and carefully explain) how these peptides are different from theirs (what does it mean 'shorter'?).

2. A detailed structural comparison should be provided analyzing the structural details of the interface with pS2 and comparing this with the extended diheptad interface presented here.

3. To support their claims on the differences with the previous structure (made several time throughout the manuscript) some additional NMR data (i.e. chemical shift perturbations) comparing pS2 and pT4 peptides should be performed to support the conclusion that the extended peptide is important also for pS2 recognition. Otherwise, the current claims are speculation.

- Related to the previous point, it appears that most of the "c" region in the peptide used is not recognized - binding affinities to a corresponding shorter peptide should be compared with the data presented.

- In the diheptaed peptide used, only one T4 is phosphorylated, while in the context of the holo CTD all T4 positions are presumably phosphorylated. What is the affinity for the peptide with both T4 positions phosphorylated? An important question is how the interaction with the holo CTD would take place. Does binding avidity play a role?

- The binding affinity of pThr4-CTD for Rtt103p is 2.5 times 'weaker' than with the pSer2-CTD in their fluorescence study. However, in their structure, the authors report a larger binding interface on Rtt103p in the case of pThr4-CTD compared to the pSer2-CTD in the previous structure. The authors should discuss the implications for the recognition of the two types of phosphorylated peptides. Can specific features in the two structures reconcile that the affinities are yet very similar?

- As possible concern of the relevance of Thr4 or Ser2 phosphorylation is that the affinity for a non-phosphorylated peptide is not much weaker (4-fold for pT4). Can this really represent a functional relevant phosphorylation? Could this be different in the context of the holo CTD?

- In page 6, authors introduced Tyr62 and His66 at the pT4-CTD binding interface on Rtt103p and suggest that the interaction is abolished. The authors must confirm that these mutations do not affect the structural integrity of Rtt103p. This can be easily done by showing 1H,15N correlation spectra of these mutants compared to the wild type CID.

- Since the 13C labeled peptides are available, authors should also report the cis-trans differences when the peptides (pT4, pS2) are bound to Rtt103p, in addition to their cis-trans analysis on free pT4- and pS2-CTD.

- The authors conclude that pS2 and pT4 in CTD are functionally degenerate, but no "functional" data are presented. Moreover, it is still possible that these two phosphorylations may distinguish different binding partners. This conclusion should be made with caution and its limitations carefully discussed.

1st Revision - authors' response

09 March 2017

Please find attached a revised version of our manuscript entitled "**Recognition of phosphorylated threonine-4 of RNA polymerase II C-terminal domain by Rtt103p**", by Jasnovidova et al. to be considered for publication in *EMBO reports* as a short report. We have revised our manuscript in accord with the referees' suggestions/comments and the editorial requirements we received from you on the 4th of January 2017.

In our revised manuscript we did not include experiments in regard with cross-reactivity of the Thr4-P and Ser2-P antibodies as it is beyond the scope of current manuscript. These experiments were done previously using ELISA and the antibodies appear to be specific – α Ser2-P binds Ser2-P and not Thr4-P, α Thr4-P binds Thr4-P and not Ser2-P (Hintermair, et al 2012; Figure 2A and 2B).

Otherwise, we have done all additional experiments and changes suggested by the Referees. These experiments include:

1. Additional “control” FA measurement for another singly phosphorylated peptide (Tyr1-P peptide; PSpYSPTSPS YSPTSPS).
2. We measured the FA data for the pT4 peptide with both T4 phosphorylated to explore the effect of avidity
3. Additional FA binding experiment for the minimal binding moiety of the CTD and Rtt103p
4. NMR data for the Rtt103p mutants to confirm their structural integrity.
5. New NMR data for ¹³C-proline-labeled peptide bound to Rtt103p.

These new data, which are summarized in Fig. 1, Fig. EV2, and Appendix Figure S1, strongly support our original conclusions.

Our point-to-point response to the referees’ comments/questions is in a separate document.

(Our responses to the reviewers’ comments are shown in italics):

Referee #1:

The yeast transcription termination factor Rtt103p has a RNA polymerase II CTD interaction domain (CID) and can bind specifically to serine 2 and threonine 4 phosphorylated heptad repeats of CTD. While the interaction of Rtt103p with CTD phosphorylated at serine 2 residues has been studied before (Lunde et al., 2010, Ref. 20) , it is currently unclear, which structural changes in CID can discriminate between Ser2P and Thr4P residues in CTD.

Stefl and coworker now provide evidence that only a minimal change in the structure of CID is responsible for this discrimination. Using NMR analysis the authors show that the arginine at position 108 can switch between two positions that contacts either the hydroxyl group of Ser2 and the phospho group of Thr4P, or, vice versa, the hydroxyl group of Thr4 and the phospho group of Ser2P. Consistent with this finding the authors show that interaction of Rtt103p interaction is blocked, if both residues are phosphorylated.

The NMR studies are complemented and confirmed by the study of Rtt103p and CTD mutants. The CTD Thr4/Glu mutant is lethal. This may be explained by the finding that the acidic glutamine residue blocks the binding of Rtt103p (and other factors) to Ser2P residues.

The authors discuss a model in which the 3' termination machinery can be recruited by different CTD phosphorylation marks and that the "CTD code" is degenerated. In an alternative (or more general) model, Ser2P and Thr4P marks may regulate termination of transcription in a gene specific manner. Testing of this model would require the analysis of CTD Thr4 mutants in transcription termination.

In summary, this well performed study explains how a CID can bind two different CTD marks just by switching the conformation of single amino acid in the binding domain.

Minor point:

page 3 line 14: in this study a T4A mutant was analysed

The referee is correct. There is a typo. Indeed, ref. no. 12 (Hintermair et al.) reported that the T4A mutant (not T4V) showed defect in transcription elongation. In the revised version of our manuscript, it now reads “In agreement with this, the T4A mutant showed defect in transcription elongation”¹²

Referee #2:

Phosphorylations on the C-terminal domain (CTD) of the largest subunit of RNA polymerase II play important roles in mediating the recruitment of RNA processing factors and chromatin modifiers during transcription. The work by Jasnovidova et al. presents an interesting new finding related to

Threonine4-phosphorylation (Thr4-P). Based on Fluorescence Anisotropy and NMR structural studies, the authors clearly show that the CTD-interacting domain (CID) of Rtt103, a well-known CTD Ser2-P binder, can also bind to Thr4-P with comparable affinity *in vitro*. Interestingly, the phosphate on either Ser2 or Thr4 occupies a nearly identical position in the structure. In addition, they demonstrate that a CTD with phosphorylations on both Ser2 and Thr4 has decreased affinity for Rtt103 CID, consistent with the structure predicting that this doubly phosphorylated state would interfere with binding. However, recent mass spec papers suggest this double phosphorylation probably doesn't exist *in vivo*. Altogether, this study sets forth an important and intriguing model where proteins that specifically recognize Ser2-P might also bind to Thr4-P and vice versa.

We thank the reviewer for the kind words and the very constructive suggestions.

Comments:

1. To argue that CID interaction with Ser2-P or Thr4-P is specific among singly phosphorylated forms, it would have been good to include another singly phosphorylated peptide in the FA experiment as a control, such as Ser7-P or Ser5-P (as done in Lunde et al. 2010; Ni et al. 2014). If the authors think that this has been established well enough in other papers that used shorter peptides, this point should at least be mentioned for clarification.

We followed the referee's suggestion and measured the FA data for another singly phosphorylated peptide (Tyr1-P peptide; PS pYSPSPS YSPSPS). Interestingly, Tyr1-P in the central heptad completely abolished the binding with Rtt103p. This new "control" data together with already available binding affinities for Ser7-P or Ser5-P complete the overview of binding affinities for various CTD phosphoisoforms.

We modified our text on page 5. "The pSer5 and pSer7 phosphorylation marks were also previously shown to abolish and lower the binding with Rtt103p, respectively²⁰⁻²¹. Next, we introduced the pTyr1 mark to the central heptad of the CTD peptide which completely abolished the binding with Rtt103p (Fig 1B). This suggests that Y_{1b} is accommodated in the hydrophobic pocket following the previously established binding model for CIDs¹⁷."

2. One concern I have about Figure 1 FA experiment is that the Thr4-P peptide signal is significantly higher than the other peptides and perhaps not even completely saturated. This could lead to an apparent K_d lower than the actual value. Can the authors comment on this?

We thank the referee for pointing this out. Indeed, the FA signal seems to be higher and less saturated for the pT4 peptide. It was also observed by others that some additional effects can appear at high protein concentration for some phosphoCTD complexes (likely due to unspecific binding contribution, increased viscosity, etc.) (Kubicek et al., 2012; Mayer et al., 2012; Sun et al., 2010). To account for these effects, we have included a non-specific binding contribution into the binding model and the fit was done using "numerical approach" (fitting differential equations; see methods) that is less sensitive to missing data points and outliers than the classical "algebraic approach".

2. It would be meaningful to add some discussion on the possibility of other CIDs (i.e. Pcf11, Nrd1, SCAF8) recognizing Thr4-P as well. As far as I understand, Pcf11 CID lacks the equivalent of Arg108 that serves as a hydrogen bond partner with Ser2-P or Thr4-P, while SCAF8 has Arg112. The authors' insights on this aspect might lead to and help follow up studies trying to understand why Pcf11 might not have this characteristic and whether Thr4-P might be relevant to SCAF8 function.

We followed the referee's suggestion and added the following discussion (page 6/7): "Akin to Rtt103p, also other CID-containing proteins such as SCAF4/8²², RPRD1A/1B/2²¹, and CHERP²¹ contain the equivalent arginine in the CID pocket. It will be interesting to see whether these human proteins really recognize pThr4-CTD as well and whether the pThr4 mark is relevant to their functions. Other CID-containing proteins in yeast, such as Nrd1p and Pcf11p, do not contain the equivalent arginine and these proteins were absent in the pThr4-CTD interactome¹⁸."

3. Regarding the difference between this structure and that from Varani group, could the different pathway of the C-terminal CTD region also be affected by the fact that Varani used a peptide where both repeats were phosphorylated at Ser2-P? Their FA experiments also showed a higher affinity for two Ser2-P repeats versus one. Could that be because they were binding in two different registers? The Greenblatt group also reported binding between RPRD1A/B CID and a 'SPS YpSPTSPS YpSPTSPS' peptide. However, their CTD peptide was only visualized down to the second Tyr, perhaps also suggesting a second Ser2-P might affect interactions with CIDs.

*We agree with the referee. We also think that a peptide where both repeats are phosphorylated at Ser2-P (YpSPTSPS YpSPTSPS) as it was used in the previous study by Lunde et al., 2010 can bind in two different registers in the CID pocket. With the knowledge of our structure and the structural work on RPRDs by Ni et al., 2014, the YpSPTSPS YpSPTSPS peptide can, in principle, bind in two different and INCOMPLETE registers, **YpSPTSPS** YpSPTSPS or YpSPTSPS **YpSPTSPS** (the binding registers are highlighted in bold). We wish to note that a multiple binding scenario often leads to difficulties for NMR data analysis.*

This is why added two additional residues at the N-terminus (PS YSPpTSPS YSPTSPS) and introduced only a single phosphothreonine residue. With this peptide, we avoided binding in two registers and obtained a high quality NMR data and we revealed that Rtt103p CID binds across three CTD heptads and that the minimal CTD binding moiety consists of the PS YSPpTSPS Y sequence in our complex. This is consistent with the RPRDs-CTD complexes (Ni et al., 2014). Adding second phosphorylation for the pT4 peptide (creates the second binding site with incomplete moiety) increases the binding affinity to Rtt103p (new Fig 1B) due to avidity (increased local concentration of binding sites), similarly as for the pS2 peptide. Furthermore, our NMR chemical shift perturbation study suggests that both pS2 and pT4 extended peptides should be accommodated in the same way (see Referee #3, comment no. 1; Figure EV1).

As requested by the Referee no.3, we added a new Figure EV3 that compares the peptide sequences used for structural calculations of the pT4 and pS2 complexes. The new Figure EV3 also compares the interfaces between Rtt103p and the two (pS2 and pT4) peptides (see below). We believe that the new figure helps to visualize the 3D structure of the interfaces and their differences.

4. p6. The paper says Arg108 is direct reader of Thr4-P, but this residue also makes a similar contact when the phosphate is on Ser2. The Asn mutation similarly blocks Ser2P peptide in FA assay. So I don't think it's accurate to say this residue is a direct reader of Thr4P, but rather only the phosphate at this location, whether or not it's on Thr4 or Ser2.

We removed our statement that Arg108 is a direct reader of pThr4 on page 6.

5. I would clarify two points in the discussion related to the recent mass spec papers from the Buratowski and Eick groups:

A. p4 and 9. Reference 11 (Suh et al) reports only a very small amount of Thr4P (far less than both Ser2P and Ser5P) in the Rtt103 pulldowns. The Schuller paper (reference 10) doesn't pull down Rtt103, but reports a higher level of Thr4P in total pol II, similar to Ser2P (although that paper also has a substantial False Location Rate). So it's unclear how much Thr4P exists in cells.

B. p5. Neither mass spec study finds very much Tyr1-P in yeast, so the proposed model (reference 17) where Tyr1-P blocks CID binding to an adjacent Ser2-P or Ser5-P is very unlikely.

Following the referee's suggestion, we discuss the uncertainty of Thr4 phosphorylation level in cells on page 3. We have also rephrased our statement on page 9.

We followed the suggestion regarding the model in which Tyr1-P prevents binding of CID-containing proteins during transcription elongation. The statement "Due to tight interaction of the Y1b residue with CIDs, it is generally assumed that phosphorylation of this tyrosine in the CTD prevents binding of CID-containing protein factors during transcription elongation phase" was removed.

6. The remarkably similar structures of the Thr4-P and Ser2-P peptides bound to Rtt103 makes me question whether the antibodies against these modifications are really as specific as advertised (the

original Eick paper doesn't show any quantitation). It seems likely there might be at least some level of cross-reactivity. I wonder if the authors have tested their CTD peptides with the antibodies. Although this might seem slightly off topic, the results would shed light on whether this is a general issue that needs to be taken into consideration for all Ser2-P and Thr4-P experiments. Given how much people rely on the antibodies, this is an important issue for the CTD field.

We thank the reviewer for this insightful comment. We agree with her/him in the notion that the cross-reactivity of antibodies could be critical for many biological assays. Fortunately, antibodies developed against different CTD phosphoisoforms do not employ a CTD-interacting domain (CID), a domain present in a number of transcription and processing factors. CIDs are rather weak binders of CTD (beneficial for a rapid re-assembly of the RNAPII transcription machinery). Specific antibodies utilize different mechanism to recognize the phosphorylated CTD and bind the CTD with K_{DS} of several orders of magnitude lower compared to CIDs. We did not test our CTD peptides with the Ser2-P and Thr4-P-specific antibodies as this was done previously with similar peptides using ELISA and the antibodies appear to be specific - aSer2-P binds Ser2-P and not Thr4-P, aThr4-P binds Thr4-P and not Ser2-P (Hintermair, et al 2012; Figure 2A and 2B). We think that probing the cross-reactivity of antibodies using another approach is beyond the scope of current manuscript.

Referee #3:

The manuscript of Jasnovidova et al reports the 'first' structure of transcription termination factor Rtt103p-CID bound to a singly Thr4-phosphorylated CTD diheptad N-terminally extended by Pro-Ser (pT4) using heteronuclear NMR spectroscopy. An increased binding affinity (> 4X) CTD for Rtt103p upon phosphorylation at the first repeat of T4 is determined by fluorescence anisotropy. The structure and binding affinities are compared with the recognition of S2 phosphorylated CTD peptides as reported by Lunde et al. The authors propose that the pT4 CTD peptide has an extended binding interface on Rtt103p compared to pS2 binding reported by Lunde et al, mainly due to the usage of longer peptides in the current study. Finally, the authors conclude that double S2 and T4 phosphorylation reduces the CID interaction due to electrostatic repulsion, while the affinities and molecular functions of single phosphorylation at S2 and T4 appear comparable and redundant. The experimental approaches and data are in general sound and the manuscript represents a short report on a structure with binding studies. Unfortunately, the discussion appears short and lacks rigor and depth, and will benefit from a more comprehensive analysis, especially regarding a comparison with the previous study on the recognition of S2 phosphorylation by the Rtt103p CID.

We thank the reviewer for her/his in-depth review and the very constructive suggestions.

Specific comments:

- In the introduction the authors refer to the previous study by Mayer et al stating that pT4 prevents binding of transcription termination factors. In contrast, the present work and other studies describe controversially that the transcription termination factor Rtt103p is recruited more efficiently with pT4 phosphorylation of the CTD. These controversial findings should be discussed.

We followed the referee's suggestion and discuss these findings on page 9. In brief, our structural data explain why the original suggestion that pT4 could interfere with binding to a CID is invalid. Akin to pS2, pT4 also forms intramolecular hydrogen bond stabilizing the bound CTD conformation, and also forms a salt bridge with arginine 108. See page 9.

- The authors argue that the previous study by Lunde et al exhibited 'incomplete binding' (bottom, p.4) due to the usage of a shorter CTD peptide. This is a speculation that should be clarified.
 1. The authors should compare (and carefully explain) how these peptides are different from theirs (what does it mean 'shorter'?).

The previous study by Lunde et al. used a peptide containing two repeats of canonical heptads (YpSPTSPS YpSPTSPS) for the structural determination of the Rtt103-CTD complex. For the pT4 peptide, we added two additional residues at the N-terminus (PS YSPpTSPS YSPTSPS) based on our preliminary data and the structural and binding data from Ni et al. 2014. Furthermore, we introduced only a single phosphorylation to avoid multiple-register binding scenario that would

complicate NMR data analyses. With this peptide, we obtained a high quality NMR data and we revealed that Rtt103-CID binds across three CTD heptads and that the minimal CTD binding moiety consists of the PS YSPpTSPS Y sequence.

In the revised version of our manuscript, we added a new Figure EV3 which compares the peptide sequences used for structural calculations of both complexes. The new Figure also compares the interfaces between Rtt103 and the two (pS2 and pT4) peptides (see below). We believe that the new figure with helps to visualize the 3D structure of the interfaces and their differences. See also response to the comment no. 3 from Referee 2.

2. A detailed structural comparison should be provided analyzing the structural details of the interface with pS2 and comparing this with the extended diheptad interface presented here.

We followed the Referee's suggestion and prepared a new side-by-side comparison of the interfaces for the pS2 and pT4 peptides bound to Rtt103. The new Figure EV 3 shows the 3D structures with interfaces.

3. To support their claims on the differences with the previous structure (made several time throughout the manuscript) some additional NMR data (i.e. chemical shift perturbations) comparing pS2 and pT4 peptides should be performed to support the conclusion that the extended peptide is important also for pS2 recognition. Otherwise, the current claims are speculation.

We apologize for not presenting our NMR data clearly enough. The recommended chemical shift perturbations comparing pS2 and pT4 peptides are shown in Figure EV1. This comparison with equivalent peptides (singly phosphorylated pSer2-CTD and pThr4-CTD peptides with complete binding moiety) shows identical chemical shift perturbations except for residues in the vicinity of phosphomarks which differs in the two peptides. In the revised version of our manuscript, we highlight more these NMR data (page 8) to support our claim.

- Related to the previous point, it appears that most of the "c" region in the peptide used is not recognized - binding affinities to a corresponding shorter peptide should be compared with the data presented.

We followed the Referee's suggestion and prepared the recommended shorter peptide that contained only the residues for which we observed experimental contacts in our NMR experiments. It binds with a K_d of 18 ± 1 . The data are now included in the revised version of our manuscript. On page 6, it reads "This minimal CTD-binding moiety binds Rtt103p CID with a K_D of $18 \pm 1 \mu M$ (assayed by FA) which is almost identical as pThr4-CTD used for structural determination."

- In the diheptaed peptide used, only one T4 is phosphorylated, while in the context of the holo CTD all T4 positions are presumably phosphorylated. What is the affinity for the peptide with both T4 positions phosphorylated? An important question is how the interaction with the holo CTD would take place. Does binding avidity play a role?

We used a single T4 phosphorylated peptide in the central "b" heptad to avoid binding in multiple registers that would complicate NMR analyses. Furthermore, the data measured for the peptide with both T4 positions phosphorylated could also suffer from the exchange processes.

To answer the Referee's question concerning the avidity, we measured the FA data for the pT4 peptide with both T4 phosphorylated. As the K_d decreased to $6 \pm 0.2 \mu M$ for this substrate, it seems that the binding avidity play a role if the CTD is phosphorylated at multiple sites due to increased local concentration of binding sites. The new data are included in Fig 1 in the revised version of our manuscript. We also modified the text on page 5 accordingly "Doubly phosphorylated pThr4-CTD at both Thr4 displayed increased binding affinity due to avidity effects ($6 \pm 0.2 \mu M$)."

- The binding affinity of pThr4-CTD for Rtt103p is 2.5 times 'weaker' than with the pSer2-CTD in their fluorescence study. However, in their structure, the authors report a larger binding interface on Rtt103p in the case of pThr4-CTD compared to the pSer2-CTD in the previous structure. The

authors should discuss the implications for the recognition of the two types of phosphorylated peptides. Can specific features in the two structures reconcile that the affinities are yet very similar?

Our data indicate that the binding area is not different for the pThr4-CTD and pSer2-CTD peptides of equivalent size, see NMR data comparison (point no. 3). The difference in binding affinity could reflect the geometry of binding and/or different chemical nature of pSer2 and pThr4 contributing to formation a more optimal salt-bridge for the pSer2 peptide than for the pThr4 peptide.

- As possible concern of the relevance of Thr4 or Ser2 phosphorylation is that the affinity for a non-phosphorylated peptide is not much weaker (4-fold for pT4). Can this really represent a functional relevant phosphorylation? Could this be different in the context of the holo CTD?

We agree with the referee and our new data show so (see above), that the avidity effect contributes to the stronger binding of phosphorylated CTD in the context of holo CTD.

- In page 6, authors introduced Tyr62 and His66 at the pT4-CTD binding interface on Rtt103p and suggest that the interaction is abolished. The authors must confirm that these mutations do not affect the structural integrity of Rtt103p. This can be easily done by showing 1H,15N correlation spectra of these mutants compared to the wild type CID.

We thank the reviewer for suggesting of this important control. We measured 1H NMR data for these two critical mutants (Tyr62Ala and His66Ala) as well as for the wild-type CID. The new data are summarized in Appendix Figure S1 and introduced on page 6. These data confirm the structural integrity of the mutants.

- Since the 13C labeled peptides are available, authors should also report the cis-trans differences when the peptides (pT4, pS2) are bound to Rtt103p, in addition to their cis-trans analysis on free pT4- and pS2-CTD.

We have followed the Referee's suggestion and extended our cis-trans analysis for the pT4 peptide. The PS YS¹³C P(pT)SPS YS peptide was titrated with Rtt103-CID and monitored by [¹H,¹³C]-HSQC. As expected, the peaks in the HSQC spectra corresponding to the cis conformation disappeared during titration, indicating a shift in the cis-trans equilibrium towards the trans conformation of the Ser-Pro prolyl-peptidyl bond that is required for the β-turn formation. Furthermore, the peaks in the HSQC spectra corresponding to the trans conformation of Pro_{3b} moved upon titration with protein, reflecting the accommodation of the proline in the hydrophobic pocket of Rtt103p CID. We added this experiment in the revised version of our manuscript (page 8/9, Fig EV2c). It reads "Next, we titrated the PS YS¹³C P(pT)SPS YS peptide with Rtt103p CID and monitored the titration by [¹H,¹³C]-HSQC experiment (Fig EV2c). The spectra show the disappearance of peaks that correspond to the cis conformation during titration, indicating a shift in the cis-trans equilibrium towards the trans conformation of the Ser_{2b}-Pro_{3b} prolyl-peptidyl bond that is required for the β-turn formation. The peaks corresponding to the trans conformation of Pro_{3b} moved upon titration with protein, reflecting the accommodation of the proline in the hydrophobic pocket of Rtt103p CID."

We could not do the same titration experiment with doubly phosphorylated peptide (pT2/pT4; PS Y(pS)¹³C P(pT)SPS YS) as the peptide degraded and the new one was not produced on time. We expect that the doubly phosphorylated peptide would manifest similar perturbations albeit of a smaller magnitude as it a weaker binder.

- The authors conclude that pS2 and pT4 in CTD are functionally degenerate, but no "functional" data are presented. Moreover, it is still possible that these two phosphorylations may distinguish different binding partners. This conclusion should be made with caution and its limitations carefully discussed.

We agree with the referee that other CTD-binding factors recognizing the CTD via different mechanism may distinguish these two phosphorylations from each other. Therefore, we have softened our statement throughout the manuscript and linked it to CID-containing proteins.

Thank you for the submission of your revised manuscript. We have now received the enclosed reports from the referees that were asked to assess it. As you will see, referee 3 only has one more minor suggestion that I would like you to incorporate before we can proceed with the official acceptance of your manuscript.

A single figure in the Appendix is not really useful, we can include this additional figure as an exceptional sixth EV figure and delete the Appendix file. Please change the figure name and the callouts in the manuscript text accordingly.

Please include the PDB codes for your structures in the materials and methods section.

When you upload a new manuscript version, you can bring forward the old files, so they do not need to be uploaded again. You can then just replace the files that need to be replaced.

We still need an ORCID ID from you. Please go to your personal profile page in our online manuscript tracking system and insert your ORCID ID. We can unfortunately not do this for you.

I would like to suggest a few minor changes to the title and abstract:

Structural insight into recognition of phosphorylated threonine-4 of RNA polymerase II C-terminal domain by Rtt103p

Phosphorylation patterns of the C-terminal domain (CTD) of RNA polymerase II (called the CTD code) orchestrate the recruitment of RNA processing and transcription factors. Recent studies showed that not only serines and tyrosines but also threonines of the CTD can be phosphorylated with a number of functional consequences, including the interaction with yeast transcription termination factor, Rtt103p. Here, we report the solution structure of the Rtt103p CTD-interacting domain (CID) bound to Thr4 phosphorylated CTD, a poorly understood letter of the CTD code. The structure reveals a direct recognition of the phospho-Thr4 mark by Rtt103p CID and extensive interactions involving residues from three repeats of the CTD heptad. Intriguingly, Rtt103p's CID binds equally well Thr4 and Ser2 phosphorylated CTD. A doubly phosphorylated CTD at Ser2 and Thr4 diminishes its binding affinity due to electrostatic repulsion. Our structural data suggest that a CID-containing CTD code reader may recognize different letters of the CTD code, suggesting redundancy in the code.

The last sentence of the abstract is not really clear to me. Can you may be rewrite it? What makes you think that the recognition of different letters indicates redundancy?

EMBO press papers are accompanied online by A) a short (1-2 sentences) summary of the findings and their significance, B) 2-3 bullet points highlighting key results and C) a synopsis image that is 550x200-400 pixels large (the height is variable). You can either show a model or key data in the synopsis image. Please note that text needs to be readable at the final size. Please send us this information along with the revised manuscript.

REFEREE REPORTS

Referee #2: The authors have addressed my concerns. It's an interesting and convincing paper that is appropriate for EMBO reports.

Referee #3: The authors have addressed the concerns raised. Concerning the use of a peptide with a single phospho site, the authors should mention that one reason for this was to improve spectral quality for structural analysis by avoiding binding in multiple registers. Beyond this minor suggestion, I recommend publication.

2nd Revision - authors' response

23 March 2017

Authors made the requested changes and submitted the final version of their manuscript.

3rd Editorial Decision

29 March 2017

I am very pleased to accept your manuscript for publication in the next available issue of EMBO reports. Thank you for your contribution to our journal.

YOU MUST COMPLETE ALL CELLS WITH A PINK BACKGROUND PLEASE NOTE THAT THIS CHECKLIST WILL BE PUBLISHED ALONGSIDE YOUR PAPER

Corresponding Author Name: Richard Stefi, Olga Jasnovidova
Journal Submitted to: EMBO reports
Manuscript Number: EMBOR-2016-43723V1